# Natural History of Stargardt Disease: The Longest Follow-Up Cohort Study

**DOI:** 10.3390/genes14071394

**Published:** 2023-07-02

**Authors:** Jana Sajovic, Andrej Meglič, Ana Fakin, Jelka Brecelj, Maja Šuštar Habjan, Marko Hawlina, Martina Jarc Vidmar

**Affiliations:** 1Eye Hospital, University Medical Centre Ljubljana, Grablovičeva ulica 46, 1000 Ljubljana, Slovenia; jana.sajovic@kclj.si (J.S.); andrej.meglic.info@gmail.com (A.M.); ana.fakin@gmail.com (A.F.); jelka.brecelj@gmail.com (J.B.); sustar.majchi@gmail.com (M.Š.H.); marko.hawlina@gmail.com (M.H.); 2Faculty of Medicine, University of Ljubljana, Vrazov trg 2, 1000 Ljubljana, Slovenia

**Keywords:** long-term follow-up, STGD1, ERG groups, Fishman stages, DDAF area, genotype-phenotype correlations

## Abstract

Long-term natural history studies are important in rare disease research. This study aimed to assess electrophysiological and fundus autofluorescence (FAF) progression rate in 18 genetically confirmed Stargardt disease (STGD1) patients with a minimum follow-up of 10 years. Age at the first and last exams, age at onset, Snellen decimal visual acuity (VA), electroretinography (ERG), and FAF images were evaluated. Patients were classified into four Fishman stages and three electroretinography groups, and areas of definitely decreased autofluorescence (DDAF) were measured. Patients were further substratified based on genotype, and phenotype-genotype correlations were performed. The median follow-up was 18 (range 10–26) years. The median yearly VA loss was 0.009 (range 0.002–0.071), while the median progression rate of the DDAF area was 0.354 (range 0.002–4.359) mm^2^ per year. Patients harbouring p.(Gly1961Glu) or p.(Asn1868Ile) allele had significantly slower DDAF area progression when compared to patients with other genotypes (0.07 mm^2^ vs. 1.03 mm^2^, respectively), as well as significantly later age at onset (20 years vs. 13 years, respectively). Results showed that structural and functional parameters, together with genotype, should be considered when counselling patients regarding prognosis and monitoring disease progression. Patients harbouring hypomorphic variants p.(Gly1961Glu) or p.(Asn1868Ile) presented with overall milder disease than patients with other genotypes.

## 1. Introduction

Stargardt disease (STGD1) is an autosomal recessive disease caused by biallelic variants in the *ABCA4* gene [1]. It is the most frequent retinal dystrophy caused by a single gene, affecting approximately 1 in 8000–10,000 individuals worldwide [2]. Currently, there are 2397 (www.lovd.nl/ABCA4, accessed on 29 June 2023) known disease-causing variants. Many of them are hypomorphic that are typically very frequent (i.e., minor allele frequency is up to 7%), manifest only under certain conditions (i.e., when in *trans* with a severe variant) and are associated with a later disease onset and a milder phenotype. The most prominent of these variants is p.(Asn1868Ile), with a minor allele frequency of 7% in the European population. Some other hypomorphic variants are, for example, p.(Gly1961Glu), p.(Arg2107His), c.769-784C>T, and c.4253+43G>A [3,4].

In addition to the extremely heterogenous genetic spectrum, there is a large phenotypic heterogeneity of STGD1 as well. The disease ranges from macular dystrophy to generalized photoreceptor dystrophy. Onset in childhood or adolescence is most common and is least frequent in later adulthood, with a worse prognosis usually associated with an earlier beginning of disease symptoms [1,5,6]. Studies suggest that a significant part of the phenotypic variability in STGD1 is related to the genotype. However, genotype-phenotype correlation studies in STGD1 are complex due to the enormous combinations of *ABCA4* variants that patients harbour [1].

Characterization of the natural history of STGD1 is vital to understanding the disease better, elucidating disease mechanisms, monitoring disease progression, and evaluating the safety and efficacy of clinical trials and treatments. Therefore, long-term studies are particularly important in rare disease research. Many retrospective and prospective studies described disease progression using quantitative assessments of retinal function and structure, as well as qualitatively by using several classifications [7,8,9,10,11,12,13,14,15,16,17]. Most commonly, Fishman classification [18] and classification based on electroretinography (ERG) characteristics (ERG groups) [19] were used.

Unfortunately, there is still no conclusive information about the natural course of STGD1. Moreover, there is still a need for reliable and sensitive quantitative and qualitative structural and functional prognostic parameters. In addition to heterogeneity, the reason might also be the lack of long-term follow-up studies, where patients would be systematically evaluated for a decade or more.

The present longitudinal study aimed to better understand the disease and assess the functional and structural changes in Slovenian patients with a clinical and genetic diagnosis of STGD1 and a median follow-up of 18 years. This study also provided the association of progression parameters with molecular genetic findings.

## 2. Materials and Methods

### 2.1. Patients

The study included 18 patients (13 females and 5 males) with a clinical and molecular diagnosis of STGD1 and a minimum follow-up duration of 10 years. The panel included 2 sibling pairs. All patients were recruited from the Slovenian registry of 1157 patients with inherited retinal diseases and examined at the Eye Hospital, University Medical Centre Ljubljana, Slovenia. The patients were further substratified based on genotype. It was previously shown that patients with p.(Asn1868Ile) and p.(Gly1961Glu) exhibit milder, often overlapping, phenotypes [4]. Therefore, we compared patients harbouring hypomorphic variants p.(Gly1961Glu) or p.(Asn1868Ile) (i.e., “hypomorphic group”) to patients with all other genotypes. As the disease was very much symmetrical between the two eyes, the right eyes were chosen for the analysis.

### 2.2. Clinical Analysis

The phenotypic analysis included age at the first and the last exams, age at onset, best-corrected visual acuity (BCVA), electroretinography (ERG), and fundus autofluorescence (FAF).

Age at onset was defined as the age at which patients first noted decreased visual acuity (VA). Disease duration was calculated from the age at the exam and the age at onset. The follow-up period was defined as the time between the first and the last exam. BCVA was measured using Snellen charts. In some patients, at their last visit, Snellen vision was determined using Tabletop Refraction System TS-610 (Nidek Co., Ltd., Gamagōri, Japan). VA at the first and last exam was evaluated and the rate of VA decline per year was calculated. A threshold equal to or below 0.1 represented legal blindness. A slit lamp fundus examination was conducted after pupil dilation.

### 2.3. Analysis of Electroretinography

ERG was used for the quantification of macular (pattern ERG; PERG) and generalized (full-field ERG; ffERG) retinal function. The recordings were made according to the standards of the International Society of Clinical Electrophysiology of Vision (ISCEV) [20,21]. The recording electrode was an HK loop placed in the fornix of the lower eyelid [22], the silver chloride reference electrode was placed on the ipsilateral temple, and the ground electrode was positioned on the forehead. First, the PERG was elicited with a 0.8° checkerboard pattern with 99% contrast and temporal frequency of 1.8 Hz, presented on a 30.7 × 23.6 (large field PERG) [23] cathode ray tube screen stimulator using an Espion (Diagnosys LLC, Lowell, MA, USA) visual electrophysiology testing systems. Two different recording systems were used to measure ffERG: Espion (Diagnosys LLC, Lowell, MA, USA) or RETI scan (Roland Consult Stasche & Finger GmbH, Brandenburg an der Havel, Germany). Rod system function was assessed with dark-adapted (DA) 0.01 ERG b−wave and DA 3 ERG a−wave amplitudes. The cone system function was tested with light-adapted (LA) 30 Hz ERG and LA 3 ERG b−wave amplitudes.

Based on ERG abnormalities, our STGD1 patients were classified into 3 groups (see Figure 1), designed by Lois et al. [19]: ERG group 1 had an abnormal macular function, ERG group 2 had an abnormal macular function and generalized cone dysfunction, and ERG group 3 presented with abnormal macular function and generalized cone and rod dysfunction.

### 2.4. Analysis of Fundus Autofluorescence Images

FAF imaging was performed using Heidelberg Spectralis (Heidelberg Engineering, Heidelberg, Germany) and Topcon retinal camera TRC-50DX (Topcon Corporation, Tokyo, Japan).

According to the appearance of FAF images, STGD1 patients were classified into 4 Fishman stages (see Figure 2) [18]: stage I was characterised by a central, atrophic-appearing macular lesion with or without flecks inside vascular arcades, stage II had numerous flecks, extending outside vascular arcades and nasally to the optic, stage III with most diffused flecks resorbed and choriocapillaris atrophy within the macula, and stage IV had diffusely resorbed flecks, atrophy of the retinal pigment epithelium (RPE) and extensive choriocapillaris atrophy throughout the posterior pole.

Images were additionally processed using our own custom-written codes in MATLAB (The MathWorks, Inc., Natick, MA, USA). As FAF images were taken with different degree objective lenses on different occasions, all images were cropped to 30°. If possible, consecutive images were aligned. The manual alignment included only rigid transformations: rotation and horizontal and vertical translation without scaling or shearing. Images were then filtered with a 2-by-2-pixel median filter to reduce noise and corrected for uneven illumination. Our approach for the determination of the definitely decreased autofluorescence (DDAF) area, representing RPE atrophy, was based on the ProgStar criteria [8]. Areas with at least 90% darkness between the healthy retina (0%) and dark reference (100%) were determined as DDAF. For the black reference point, the main blood vessel near the optic disc or optic disc was chosen. Choosing the reference point for a healthy retina was, however, more challenging. In some cases, we had to use optical coherence tomography (OCT) images to help us find the unaffected part of the retina. In cases with almost complete atrophy on 30° images, we found the grey reference point on 55° images. As fundus abnormalities increase with age, the positions of both reference points were determined on the latest image and, if possible, the same positions were chosen on all prior images. The optic disc and blood vessels were excluded manually. Black pixels were then automatically summated and the area was converted into square millimetres (mm^2^). The internal reference was used for the determination of single-image pixel size in most cases. When not, we used a scale bar on the image. The described image analysis algorithm enabled us to measure even the smallest DDAF areas on the whole FAF image, even in cases with uneven illumination of the periphery. The smallest area we could detect was defined by single-image pixel size and was on average 138 µm^2^. DDAF area at the first exam and DDAF area at the last exam were analysed, and the rate of DDAF area progression per year was calculated.

### 2.5. Genetic Analysis

Peripheral venous blood samples were obtained, and genomic DNA was extracted from blood samples according to the standard procedure. Sequencing of the *ABCA4* gene in 3 patients was performed using Illumina Nextera Coding Exome capture protocol, with subsequent sequencing on Illumina NextSeq550 (Illumina, San Diego, CA, USA). In 9 patients, sequencing of the entire *ABCA4* genomic locus was performed using single-molecule molecular inversion probes (smMIPs) library preparation, and the Illumina NextSeq500 sequencing platform [24]. Six patients were screened by the ABCR400 chip [25]. The variants’ segregation with the disease in available families was analysed by Sanger sequencing.

### 2.6. Statistical Analysis

Data were analysed using IBM SPSS Statistics software version 27.0 (IBM Corp. Armonk, NY, USA). Mann—Whitney U Test was applied to compare selected parameters between the first and the last exams and for genotype-phenotype analysis, where parameters between different genotype groups were studied. *p* < 0.05 was considered to indicate statistical significance.

## 3. Results

### 3.1. Clinical Findings

The clinical characteristics of patients are presented in Appendix A. The median age at the first visit was 22 (range 7–46) years, and the median age at the last visit was 40 (range 17–72) years. The median age at onset of symptoms was 16 (range 7–46) years, with a median follow-up of 18 (range 10–26) years. About 7/18 (39%) patients were observed for 20 years or more. The duration between the onset and the first exam was 1.5 (range 0–30) years. The median Snellen decimal BCVA at the first exam was 0.25 (range 0.10–0.8) and 0.04 (range 0.02–0.30) at the last exam. The median DDAF area at the first exam was 0.38 (range 0.00–13.70) mm^2^, and at the last exam, it was 8.80 (range 0.03–78.61) mm^2^. At baseline, patients had significantly better VA (*p* < 0.001) and smaller DDAF area (*p* = 0.002) than at the end of our study.

The median yearly loss of VA was 0.009 (range 0.002–0.071), while the median increase of the DDAF area was 0.354 mm^2^ (range 0.002–4.359 mm^2^) per year. During the time of the follow-up, we found out that VA deteriorated in 17/18 patients. The worsening of the VA and DDAF area with time is shown in Figure 3, Figure 4 and Figure 5.

To evaluate whether we can predict legal blindness (VA ≤ 0.1) from RPE atrophy, the relationship between longitudinal VA and DDAF area data for each patient was studied. We showed that a DDAF area of 12 mm^2^ or more indicated legal blindness in most patients (Figure 6).

### 3.2. Electrophysiological and Fundus Autofluorescence Progression

Graphical representations of ERG progression and progression of FAF changes are available in Appendix A.

To compare the progression of structural and functional changes over the follow-up period, we analysed the representation of Fishman stages within ERG groups at the first and last exams (Figure 7). At baseline, three patients with no ERG abnormalities belonged to Fishman stage I (flecks inside vascular arcades) and two to Fishman stage II (flecks outside vascular arcades). Five patients in ERG group 1 (abnormal macular function) were in Fishman stage I and one in Fishman stage II. Of four patients in the ERG group 2, (abnormal macular function with generalized cone dysfunction), one patient was in Fishman stage I, one in Fishman stage II, and two in Fishman stage III (resorbed flecks and macular choriocapillaris atrophy). Two patients in ERG group 3 (with abnormal macular function with generalized cone and rod dysfunction) were in Fishman Stage II, and one patient was in Fishman stage III. No patients were in Fishman stage IV.

At the end of our follow-up, three patients had normal ERG. Two of them belonged to Fishman stage I and one to Fishman stage III. Two patients in ERG group 1 were in Fishman stage I and the other two patients in Fishman stages II and III. All four patients in ERG group 2 belonged to Fishman stage III, whereas two patients in ERG group 3 were in Fishman stages I and II, four patients in Fishman stage III, and one patient in ERG group 3 belonged to Fishman stage IV (diffusely resorbed flecks and extensive choriocapillaris atrophy). This is the patient with generalized photoreceptor dysfunction and the worst retinal structure.

The progression of structural and functional changes in representative patients is shown in Figure 8 and Figure 9. When using Fishman classification to measure structural progression and ERG classification to measure functional progression, we found out that in two patients, FAF stayed stable while ERG progressed (e.g., Figure 8: Patient 3). In contrast, in four patients, ERG stayed stable and FAF progressed (e.g., Figure 8: Patient 8). Six patients showed progression of ERG as well as FAF (e.g., Figure 8: Patient 9 and Figure 9). Interestingly, there were six patients whose ERG and FAF were stable over the follow-up period of 18 years (Figure 8: Patient 10).

Two patients had the longest follow-up of 26 years (Appendix A: Patients 1 and 12). They are harbouring p.(Gly1961Glu) or p.(Asn1868Ile) in *trans* with a severe allele. Both showed VA decline, as well as structural and functional progression. However, none of them reached the final ERG group 3 or Fishman stage IV. An example of Patient 1 is presented in Figure 9.

### 3.3. Genotype-Phenotype Correlations

All patients had genetically confirmed biallelic *ABCA4* variants. The most common variant in our cohort was p.(Asn1868Ile), present in 9/18 (50%) patients. In three patients, it was present in *trans* with a severe allele, while six patients had it in *cis* with other variants (i.e., c.5461-10T>C, p.(Thr1726Aspfs*61), p.(Pro640Ala)). Variant p.(Gly1961Glu) was present in 5/18 (28%) patients, p.(Arg681*) in 4/18 (22%) patients, c.5714+5G>A in 4/18 (22%) patients, p.(Thr1726Aspfs*61) in 3/18 (17%) patients, and c.5461-10T>C in 3/18 (17%). For a representation of other variants, see Appendix A.

Patients with the p.(Gly1961Glu) or p.(Asn1868Ile) allele had a significantly later age at onset and, therefore, later age at exams than patients with other genotypes. The median age at the onset of three patients harbouring p.(Asn1868Ile) was 20 (range 20–30) years, while the age at the onset of five patients with p.(Gly1961Glu) was 19 (range 7–46) years. The duration of the follow-up was comparable between the two groups. The DDAF area at the first exam did not differ significantly. However, at the last exam, the DDAF area in the hypomorphic group was significantly smaller, leading to a significantly slower yearly rate of DDAF area progression. Contrary, VA progression per year did not differ significantly between the two groups. For more information see Table 1.

Bull’s eye maculopathy (BEM) phenotype was found in all patients with p.(Gly1961Glu) or p.(Asn1868Ile) allele (e.g., Figure 8: Patients 3, 9, 10 and Figure 9), while none of the patients with other genotypes exhibited BEM lesions (e.g., Figure 8: Patient 8.) No patients with hypomorphic variants progressed into Fishman stage IV and only one progressed into ERG group 2 and one into ERG group 3 (see Appendix A). Of six patients with stable ERG group and Fishman stage, four had p.(Gly1961Glu) or p.(Asn1868Ile) allele.

Patients 14, 15, and 16 with two null alleles (belonging to the group of patients with other genotypes) had the most severe clinical presentation (see Figure 3 and Figure 4); however, the group was too small to make reliable genotype-phenotype correlations. This also applies to other patients with specific genotypes (from the group of patients with other genotypes), as there were too many combinations of different variants.

### 3.4. Analysis of Siblings

The first sibling pair (Patients 6 and 7) were carrying c.5714+5G>A and p.(Thr1726Aspfs*61)(;)(Asn1868Ile). Patient 6 had the age at onset of 18 years, while the onset in Patient 7 was at 25 years of age. They were both followed for 11 years. They were in the same ERG group and Fishman stage at baseline. However, at the last exam, only ERG in Patient 7 progressed from group 2 to group 3. Initial VA was better in Patient 7 and the follow-up showed that her yearly VA deterioration was 2.1 times greater compared with Patient 6. Initial RPE atrophy (measured by DDAF area) was very similar between the sisters; however, the DDAF area progression per year was 4.7 times greater in Patient 7.

That degree of discordance was not found between Patient 14 and Patient 15, representing the second sibling pair. They were homozygous for p.(Arg2149*). In Patient 14 the age at onset and duration of follow-up commenced two years earlier than in Patient 15. At the first exam, Patient 14 was already in the final ERG group 3, while, according to fundus appearance, she was only in Fishman stage II, which did not progress over the follow-up time. At the first exam, Patient 15 was in ERG group 2 and Fishman stage II, which both progressed for one group/stage at the last exam. VA at baseline was 0.1 in Patient 14, while in Patient 15, it was 0.3. The yearly VA deterioration in Patient 14 was 3 times lesser than in Patient 15. The baseline DDAF area in Patient 14 was 0.03 mm^2^ and 0.91 mm^2^ in Patient 15, whose yearly DDAF area progression was 1.4 times greater.

## 4. Discussion

To our knowledge, this is the first study that systematically analysed STGD1 patients with a very long follow-up, enabling us to better understand disease evolvement and identify factors that correlate with disease progression. In our study, disease progression was characterized by VA, qualitative evaluation of ERG, and qualitative and quantitative assessment of FAF images. Results show that ERG attributes and fundus abnormalities complement one another and that genotype significantly affects progression rates.

The median follow-up of our group of patients was 18 years, with the longest follow-up of 26 years in Patient 1 (Figure 9) and Patient 12. In 7/18 (39%) of patients, the follow-up was 20 or more years. Even though there are descriptions of individual cases with a follow-up longer than 20 years [17,26,27,28], all other published cohort studies on STGD1 patients had a shorter duration of follow-up, which ranged between 1 to 11 years [7,8,9,11,13,14,15,16,17,26,27,28,29,30].

### 4.1. Electrophysiological and Fundus Autofluorescence (FAF) Progression Rate

As it is still unclear whether the disease occurs primarily in the RPE, photoreceptors or both [31,32,33,34,35], it is necessary to address structural and functional changes to determine the patient prognosis. In 2/18 (11%) of our patients, FAF stayed stable while ERG dysfunction progressed. Approximately 4/18 (22%) of them showed stable ERG and FAF progression. About 6/18 (33%) of patients represented simultaneous ERG and FAF progression, while in 33%, both parameters remained stable. Patients without progression had a median follow-up of 18 years, half with a follow-up of 20 or more years.

Classifications normally give robust and quick information on disease progression, but a year-to-year deterioration is lost, which is why quantitative analysis of functional and structural changes has become indispensable. We could not perform a quantitative analysis of ERG parameters as the protocols and measuring systems differed between the first and last exams. However, in a cross-sectional study, we defined biomarkers for electrophysiological assessment of subgroups of patients with STGD1, which could be used for disease progression assessment [36].

### 4.2. Progression of Definitely Decreased Autofluorescence Area

A measurement of RPE loss by determining the DDAF area is also crucial for monitoring STGD1 disease progression, as well as for the prediction of VA and legal blindness, as demonstrated in Figure 6. In this study, areas of DDAF from 30° FAF images were measured using our custom-written codes, which made it possible to detect the smallest DDAF areas, DDAF areas at the edge of the image periphery, and also DDAF areas on nonhomogeneous FAF images. Comparisons of our methods and results with other studies are challenging owing to different applications of statistical methods, cohort selection, inclusion and exclusion criteria, lack of genotype-phenotype correlations, distinct image acquisition, and processing and analysis. Other studies did not consider DDAF areas in the periphery and very small DDAF areas [8,9,15,29,37,38,39,40,41], even though they contribute to the understanding of STGD1 progression. In addition, in most reports, DDAF lesions were selected manually and subjectively [8,9,29,37,39,41], which might cause some unintentional human errors.

In our study, the DDAF area at baseline was 0.380 mm^2^ with 0.354 mm^2^ DDAF area progression per year. The baseline DDAF area in other studies was bigger [8,9,15,29,37,39,40], which means that our patients were included in the study very early after experiencing the first visual problems (i.e., 1.5 years) when the structural changes were still discrete. Four patients were included as young children less than 10 years of age. Moreover, the rate of RPE atrophy progression, measured by the DDAF area, was much faster in other studies when compared with ours. A wide range of growth rates was suggested, from 0.45 mm^2^ [15] per year to 1.58 mm^2^ per year [41] The differences in RPE atrophy progression rate might be due to the specific structure of patients in the cohorts and their varying entry time in the study, as the lesions might not grow at the same rate all the time. Previous studies with shorter follow-ups suggested that the progression rate depends on initial lesion size, meaning the bigger the initial lesion, the faster the progression [8,9,15,37,38,39,42]. A possible explanation, as the lesions in STGD1 expand centrifugally [43], might be that a larger initial lesion has a larger surface area for further growth and, therefore, a faster rate of progression [8]. On the contrary, our study did not show this correlation. When we stratified our cohort into two groups, a hypomorphic group and a group of patients with other genotypes, we found that the baseline DDAF area was a match between the two groups. This is even more valuable information when knowing that patients from both groups were examined after a comparable duration of the disease. The DDAF area at the last exam and growth rate were significantly different between the two groups, with a slower progression rate in patients with p.(Gly1961Glu) or p.(Asn1868Ile) allele. Therefore, we concluded that the progression rate varied even among patients with similar initial DDAF areas, meaning that the initial lesion does not unequivocally indicate progression and should not be taken as the only predictor of the growth rate. Considering our analysis, we propose that genotype should always be considered when predicting the disease progression and prognosis.

### 4.3. Visual Acuity Decline

Over a median period of 18 years, VA in our cohort of patients significantly declined, which is in accordance with previous studies with shorter follow-ups [16,39,44]. However, as it decayed very slowly (i.e., 0.009/year), we found that it was not sensitive enough to show a clinically relevant progression of the disease. In the course of follow-up, VA improved in only one patient (i.e., Patient 11), which might be due to improved fixation, optimal prescription glasses, test variability, and patient performance [44]. We did not notice any differences in VA progression rate between patients with p.(Gly1961Glu) or p.(Asn1868Ile) allele and patients with other genotypes, suggesting that genotype does not influence VA decline rate. Moreover, we found that patients in the hypomorphic group had an onset 7 years later but did not show significantly better VA. This is in contrast with other studies, where younger age of disease onset was associated with worse VA [13,16,44].

### 4.4. Disease Course in Patients Harbouring p.(Gly1961Glu) or p.(Asn1868Ile) Allele

It is known that patients harbouring p.(Gly1961Glu) or p.(Asn1868Ile) allele share some common clinical characteristics and present with a milder disease phenotype than patients carrying other alleles [1,4,45,46,47]. This was also confirmed by our study. Nevertheless, according to Fishman and ERG classifications, patients with p.(Gly1961Glu) or p.(Asn1868Ile) exhibit variable disease progression. Of eight patients in the hypomorphic group, four stayed in the same ERG group and Fishman stage from the beginning to the last exam (e.g., Figure 8: Patient10). In three patients, both the ERG group and Fishman stage progressed (e.g., Figure 8: Patient 9 and Figure 9), while in one patient, only the ERG group progressed (e.g., Figure 8: Patient 3).

A typical feature of patients with p.(Gly1961Glu) is BEM, which is otherwise present in around 20% of all STGD1 patients [48]. All our hypomorphic patients had BEM, which was not expressed in any patients with other genotypes. None of our patients harbouring p.(Gly1961Glu) or p.(Asn1868Ile) showed progression into Fishman stage IV and seemed to have preserved generalized photoreceptor function, as only two patients progressed into ERG group 2 or 3. Therefore, our long follow-up observations confirmed cross-sectional reports that described the limitation of the disease to the macula in patients with p.(Gly1961Glu) or p.(Asn1868Ile) [1,46]. Even though foveal sparing was reported in 85% of cases harbouring p.(Asn1868Ile) [4], all our cases with this allele had foveal atrophy. Age at onset in our patients with p.(Asn1868Ile) was earlier than described before [4,45], while age at onset for patients with p.(Gly1961Glu) was in line with other studies [46,47,48].

### 4.5. Variable Disease Courses among Siblings

Especially challenging is predicting patient prognosis in phenotypically discordant siblings carrying the same genetic variants in the *ABCA4* [26,49,50,51]. Patient 6 and Patient 7 were intriguingly different in the DDAF area progression rate, which was almost 5 times faster in Patient 7 than in her sister. VA decline was also greater in Patient 7, with the age at onset 7 years later than in Patient 6. In contrast, they had similar initial DDAF area, FAF appearance, disease duration, and similarly affected retinal function. Differences in functional visual outcomes and DDAF area between siblings with STGD1 were already observed by Valkenburg et al. [26] and Heath Jeffery et al. [51]. This is interesting to note and suggests a role of potential environmental factors [52,53] and modifier variants in and outside the *ABCA4* locus [3,54,55,56].

### 4.6. Study Strengths and Limitations

The main strength of our longitudinal study was a very long follow-up of STGD1 patients, as only the patients with a follow-up of 10 or more years were included. Some of them were included as young children. In addition, DDAF areas on 30° FAF images were quantitatively measured using custom-written codes, which made it possible to detect year–to–year deterioration. Another strength was the correlation of specific genotypes with progression parameters.

This study also has potential limitations. OCT imaging technique was not accessible when patients were first examined, which is why we did not use it for additional retinal structure analysis and, therefore, were not able to assess photoreceptor impairment. Even though it is one of the main problems in inherited retinal disease research, the second limitation was the small cohort number with the heterogeneous genetic structure, limiting the analysis of genotype-phenotype correlations. Larger multicentric longitudinal studies with a follow-up of 10 or more years, including patients with specific *ABAC4* genotypes and more structural and functional parameters analysed, would best describe the progression and substantiate our findings.

## 5. Conclusions

In summary, the study provided a median follow-up of 18 years and determined the importance of very long-term natural history studies in STGD1 patients. We shared fundamental, sensitive, and reliable information on disease progression needed for counselling patients regarding prognosis and the selection of appropriate and efficient endpoints, which are necessary for clinical trial design, as well as treatment safety and efficacy evaluation. We showed that structural and functional parameters should be addressed together as they complement each other. Moreover, genotype should be considered an important prognostic parameter, as patients harbouring hypomorphic variants p.(Gly1961Glu) or p.(Asn1868Ile) were presented with overall milder disease and slower progression.

## Figures and Tables

**Figure 1 genes-14-01394-f001:**
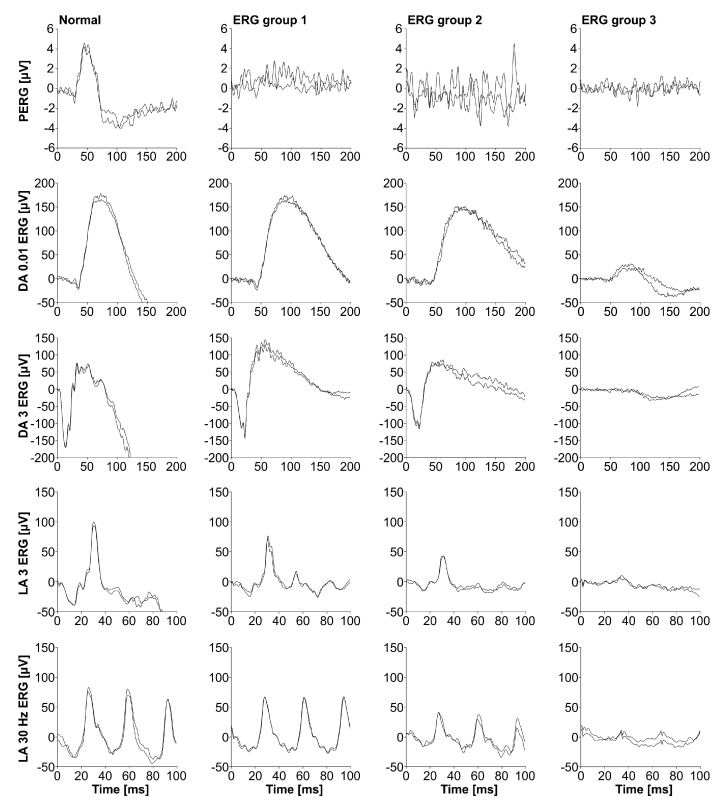
Classification based on electroretinography (ERG). Examples of ERG group 1 (Appendix A: Patient 1), ERG group 2 (Appendix A: Patient 5) and ERG group 3 (Appendix A: Patient 2) are shown. Pattern ERG (PERG) represents the macular function, dark-adapted (DA) 0.01 ERG b−wave and DA 3 ERG a−wave amplitudes represent the rod system function, and light-adapted (LA) 30 Hz ERG and LA 3 ERG b−wave amplitudes represent the cone system function.

**Figure 2 genes-14-01394-f002:**
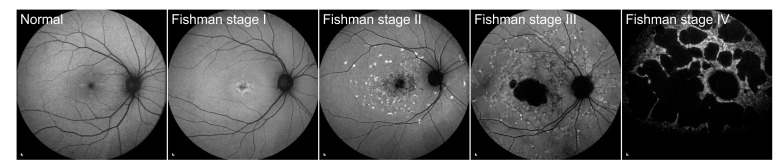
Fishman classification. Examples of Fishman stage I (Appendix A: Patient 11), Fishman stage II (Appendix A: Patient 3), Fishman stage III (Appendix A: Patient 1) and Fishman stage IV (Appendix A: Patient 2) are shown. Scale bars: 200 µm.

**Figure 3 genes-14-01394-f003:**
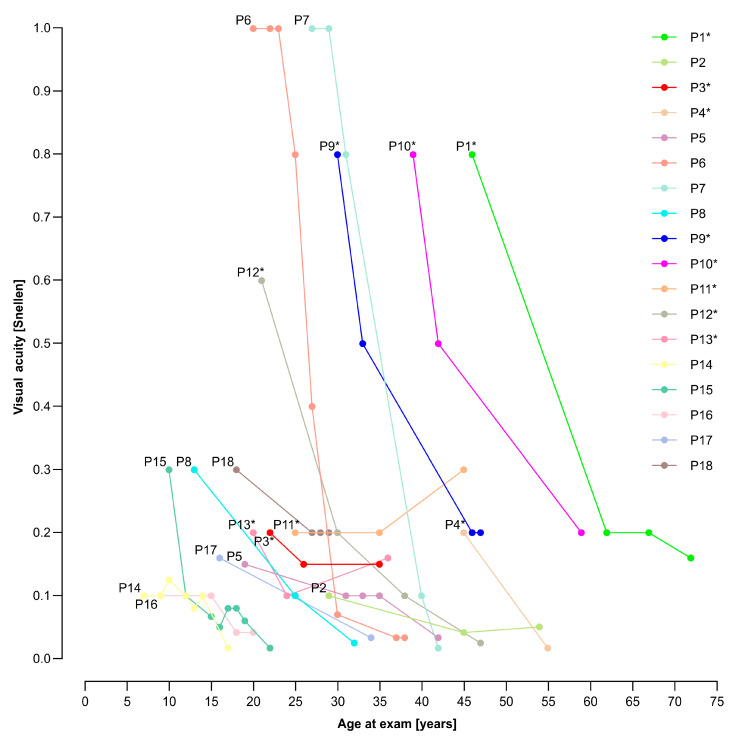
Visual acuity (VA) in relation to the age at the exam. Patients with p.(Gly1961Glu) or p.(Asn1868Ile) allele are marked with a star (*). VA declined in all patients except Patient 11. Abbreviation explanation: P—patient.

**Figure 4 genes-14-01394-f004:**
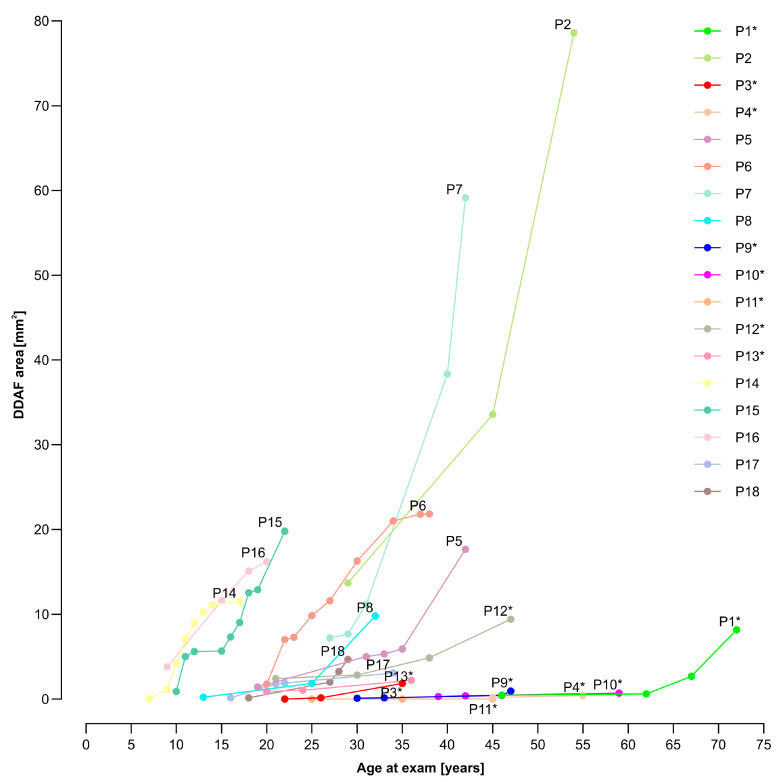
Definitely decreased autofluorescence (DDAF) area, representing retinal pigment epithelium (RPE) atrophy, in relation to age at the exam. Over the follow-up, patients with p.(Gly1961Glu) or p.(Asn1868Ile) allele had a very slow progression (i.e., P1, P3, P4, P9–P13) of retinal RPE atrophy, while others showed a very steep progression (i.e., P2, P5–P8, P14–P18). Patients with p.(Gly1961Glu) or p.(Asn1868Ile) allele are marked with a star (*).

**Figure 5 genes-14-01394-f005:**
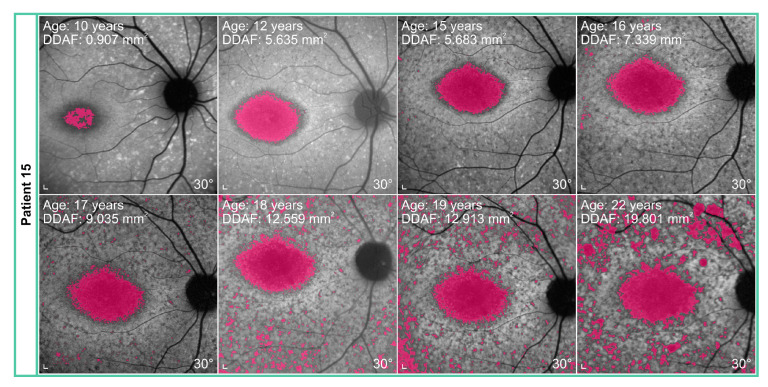
Increase in the DDAF area, shown in pink, in Patient 15 over the period of 12 years. Note the progression of the DDAF area in the image periphery. Scale bars: 200 µm.

**Figure 6 genes-14-01394-f006:**
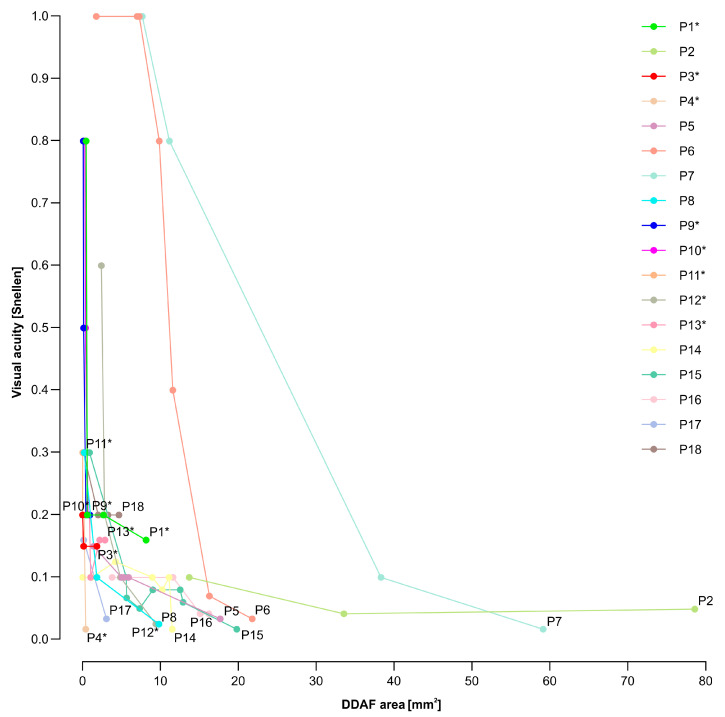
Longitudinal analysis of a relationship between VA and DDAF area. A threshold equal to or below 0.1 represented legal blindness. Patients with p.(Gly1961Glu) or p.(Asn1868Ile) allele are marked with a star (*).

**Figure 7 genes-14-01394-f007:**
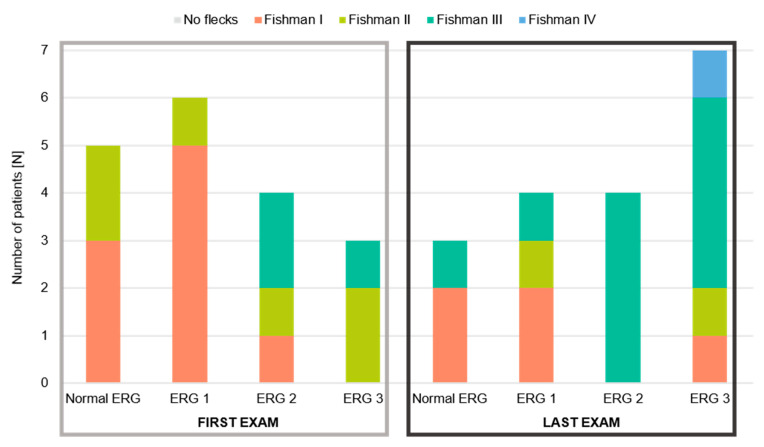
Representation of Fishman stages within ERG groups at the first and last exams. The first exams are shown within the light grey box, while the last exams are within the dark grey box.

**Figure 8 genes-14-01394-f008:**
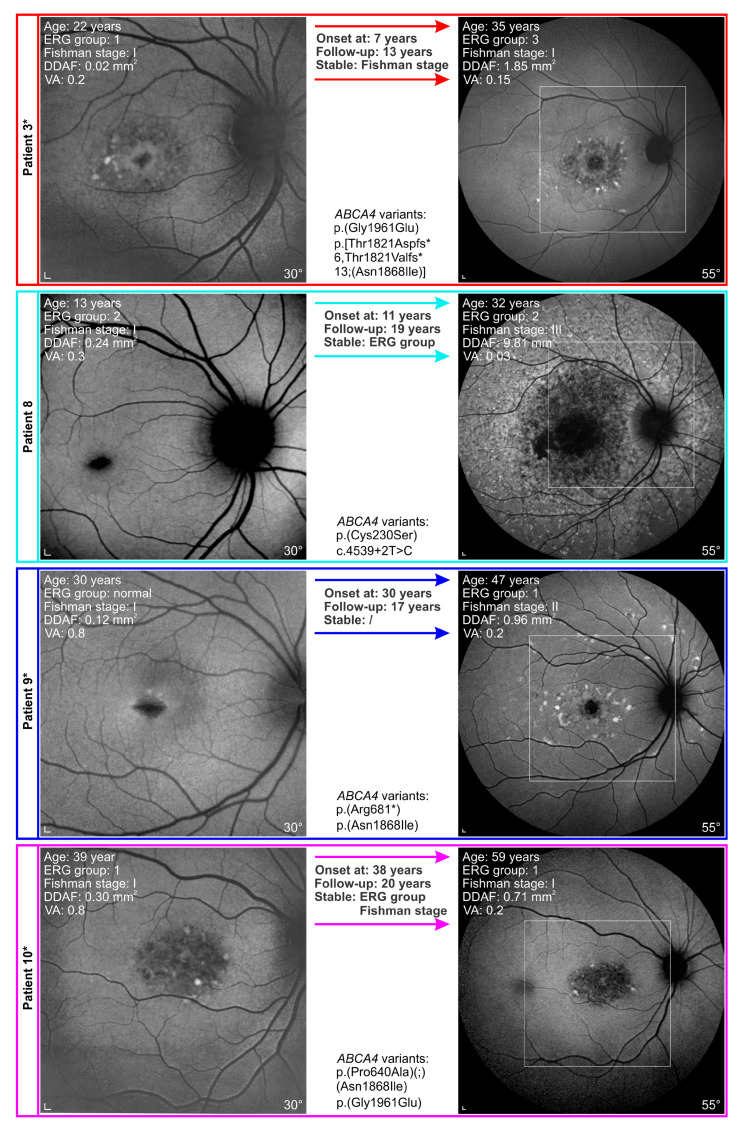
Distinct progression of functional and structural changes in Stargardt disease (STGD1) patients. Patient 3: An example of a patient with a stable fundus autofluorescence (FAF) appearance. Electrophysiologically, the patient presented with abnormal macular function at the beginning, and at the last exam, the patient progressed into ERG group 3. Patient 8: An example of a patient without flecks and almost normal FAF, while ERG analysis already showed abnormal macular function and generalised cone dysfunction at the first exam. According to ERG, rods were still preserved at the last exam, and the patient stayed in ERG group 2. On the other hand, the FAF image showed major progression into Fishman stage III. Patient 9: An example of a patient with a follow-up of 16 years who progressed from Fishman stage I to stage II. ERG also showed progression from normal ERG to ERG group 1. Patient 10: An example of a patient with no structural and functional changes of the retina described by Fishman and ERG classifications over the period of 20 years. Interestingly VA majorly declined from 0.8 to 0.2, and RPE atrophy increased from 0.30 mm^2^ to 0.71 mm^2^. Note the different progression of FAF changes between Patients 3, 9, and 10 harbouring p.(Gly1961Glu) or p.(Asn1868Ile) and Patient 8 with p.(Cys230Ser) and c.4539+2T>C variants. Patients with p.(Gly1961Glu) or p.(Asn1868Ile) allele are marked with a star (*). A white square on 55° FAF images indicates a 30° area from the first exam. Scale bars: 200 µm.

**Figure 9 genes-14-01394-f009:**
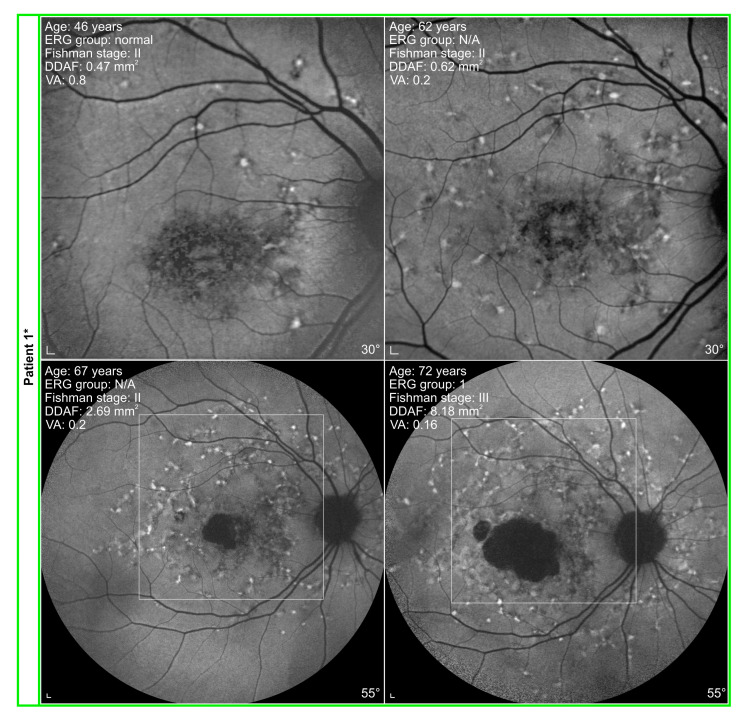
Patient with a follow-up of 26 years, harbouring p.(Gly1961Glu) in *trans* with p.(Gln1412*). Pathological fundus changes and ERG progressed with time. The absence of DDAF in the fovea at the age of 46 years might indicate the presence of foveal sparing. A white square on 55° FAF images indicates a 30° area from the first two exams. Scale bars: 200 µm.

**Table 1 genes-14-01394-t001:** Differences between Hypomorphic Group and Other Patients.

Parameter	Hypomorphic Group (n = 8) (Median, Range)	Other Patients (n = 10) (Median, Range)	Mann–Whitney Test (*p*-Value)
Age at the first exam [years]	28 (20–46)	17 (7–31)	*p* = 0.016
Age at the last exam [years]	47 (35–72)	33 (17–54)	*p* = 0.009
Age at onset [years]	20 (7–46)	13 (7–25)	*p* = 0.043
Duration of follow-up [years]	20 (13–26)	12 (10–25)	*p* = 0.055
The time between onset and the first exam [years]	1 (0–30)	3 (0–14)	*p* = 0.633
DDAF area at the first exam [mm^2^]	0.25 (0.00–2.43)	1.17 (0.03–13.70)	*p* = 0.122
DDAF area at the last exam [mm^2^]	1.41 (0.03–9.43)	16.96 (3.09–78.61)	*p* = 0.001
The yearly rate of DDAF area increase [mm^2^]	0.07 (0.0017–0.30)	1.03 (0.16–4.36)	*p* < 0.001
VA at the first exam [Snellen]	0.40 (0.20–0.80)	0.23 (0.10–0.80)	*p* = 0.122
VA at the last exam [Snellen]	0.16 (0.02–0.30)	0.03 (0.02–0.20)	*p* = 0.068
The yearly rate of VA loss [Snellen]	0.02 (0.0025–0.035)	0.01 (0.0020–0.07)	*p* = 0.740

Statistical significance is defined as a *p*-value < 0.05.

## Data Availability

The data that support the findings of this study are available upon request from the corresponding author M.J.V. or the first author J.S. The data are not publicly available due to personal data protection policies.

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
