# Peer review of "Natural History of Stargardt Disease: The Longest Follow-Up Cohort Study"

_genes, 2023, doi:10.3390/genes14071394_

Round 1
Reviewer 1 Report
The authors merit more than one compliment for this very informative and well documented study! At first, I wondered why only the RE was evaluated, but after reading the complicated processing of the FAF-pictures I understood. Unfortunately, I was not able to see the supplemental data, as the link repeatedly said "error 404- file not found". As a minor suggestion, I propose to change the colours of some patients in the figures (e.g. patient 2,7,14 and 16), as the very light colours may be difficult to perceive in the final article.
Reviewer 2 Report
The study of Sajovic et al. is a long-term longitudinal study on 18 patients with Stargardt disease from Slovenia. The minimum follow-up was 10 years. A range of parameters has been assessed to address the question regarding the disease progression over the years: electrophysiological and fundus autofluorescence (FAF) age at first and last exams, age at onset, Snellen decimal visual acuity (VA), electroretinography (ERG) and areas of definitely decreased autofluorescence (DDAF). Patients were classified into Fishman stages and electroretinography groups and assessed according to the genotype. Further, phenotypes and disease progression between siblings carrying the same variants have been compared. The illustrations facilitate the understanding of the disease progression. The limitations of the study are listed. The study is a valuable contribution to understand the disease progression in Stargardt disease and in patients carrying different genetic variants.
Comments:
1. I would add in the introduction nformation about the hypomorphic alleles, their impact on the phenotypes, and their frequency in Stargardt disease.
2. Please mention how long was the follow-up period in other studies
3. Are there any other studies on siblings with the same variant and discordant phenotypes? What could be the reasons for it?
Reviewer 3 Report
Sajovic et al. realized a very interesting article describing the “Natural History of Stargardt Disease: the Longest Follow-up Cohort Study”. I consider the manuscript very interesting but, at the same time, I suggest several revisions needed to improve the reliability and the completeness of the paper:
1. Clarification and simplification: There are some technical terms and complex sentences that could be simplified for improved readability. Consider providing definitions for technical terms or acronyms when they are first introduced (e.g. RPE, DDAF, VA, etc.). For example, instead of "Moreover, the rate of DDAF progression...", you might say "Moreover, the rate at which this retinal degeneration progresses, measured as DDAF area...".
2. Structure and Flow: Ensure that each paragraph has a clear main idea, and that ideas flow logically from one to the next. For instance, in paragraph starting with "Previous studies with shorter follow-ups suggested..." a clear topic sentence might help to guide the reader.
3. Cohesion of Sections: The transitions between sections should be smoothed out. When transitioning from discussion on "RPE atrophy progression rate" to "VA decline rate", a sentence linking these two aspects could help to guide the reader through your argument.
4. Data Presentation: Consider introducing tables or graphs to summarize complex data, such as the genotype-specific rates of disease progression. Visual representations of your findings can make them more comprehensible and impactful.
5. In-Text Citations: Ensure the numbering of your references is consistent and correct. You might also want to check the latest guidelines for the journal you're submitting to and make sure your in-text citation style matches those. Moreover, I suggest adding data related to recent bulk transcriptomics studies investigating the vascular alteration impact on several pathologies, like IRDs and MAV/CCM, which present a strong involvement of many pathways cited by the authors in the presented paper. The recent PMID: 26122242 and PMID: 36490268 could represent a substrate able to enforce the role of considered cellular mechanisms.
6. Language: The manuscript could benefit from a thorough proofreading. For example, in "Therefore, we concluded that the progression rate for a similar initial DDAF area was different...", the term "was different" is vague. Consider revising to a more precise description: "Therefore, we concluded that the progression rate varied even among patients with similar initial DDAF areas...".
7. Limitations: When discussing the limitations, explicitly mention how they might have affected your results and how future research could overcome them. For instance, explain how not having OCT imaging data from the initial exams might have influenced your findings and suggest solutions for future studies.
8. Conclusions: While the conclusion summarizes the results well, you might want to highlight the implications of your research more. For instance, what do your findings mean for the treatment of STGD1 patients? How might they inform future research?
These are just a few suggestions to improve the technical aspects of your manuscript. The content seems well-researched and detailed, and these changes could enhance its readability and impact.
The manuscript could benefit from a thorough proofreading. For example, in "Therefore, we concluded that the progression rate for a similar initial DDAF area was different...", the term "was different" is vague. Consider revising to a more precise description: "Therefore, we concluded that the progression rate varied even among patients with similar initial DDAF areas...".
